# On the Use of Convolutional Auto-Encoder for Incremental Classifier Learning in Context Aware Advertisement

## Abstract

Context Aware Advertisement (CAA) is a type of advertisement appearing on websites or mobile apps. The advertisement is targeted on specific group of users and/or the content displayed on the websites or apps. This paper focuses on classifying images displayed on the websites by incremental learning classifier with Deep Convolutional Neural Network (DCNN) especially for Context Aware Advertisement (CAA) framework. Incrementally learning new knowledge with DCNN leads to catastrophic forgetting as previously stored information is replaced with new information. To prevent catastrophic forgetting, part of previously learned knowledge should be stored for the life time of incremental classifier. Storing information for life time involves privacy and legal concerns especially in context aware advertising framework. Here, we propose an incremental classifier learning method which addresses privacy and legal concerns while taking care of catastrophic forgetting problem. We conduct experiments on different datasets including CIFAR-100. Experimental results show that proposed system achieves relatively high performance compared to the state-of-the-art incremental learning methods.

## 1 Introduction

Recently, deep neural networks (DNNs) have shown remarkable performance in several classification tasks in many domains such as speech, image, natural language processing, etc.,. Among deep networks, Deep Convolutional Neural Networks (DCNNs) are the most successful models for supervised image classification and set the state-of-the-art in many benchmarks competitions Lecun et al. (1998), Krizhevsky (2009) where the numbers of image categories have already been defined. It is common that machine learning systems with neural networks such as DCNN are trained in batch mode in which training samples from all the classes are available Wu et al.. Hence, DCNN based learning framework fits well with the requirements of benchmark evaluations.

However, in real-world applications, more and more image categories will be available as time goes on and network will need to learn new tasks. The drawback of DCNN is inability to learn new information without redoing the whole learning process using all old and new information. If previously learned information is not available at the time of adding new information, DCNN tends to forget old memories. This is referred to as catastrophic forgetting problem.

Human brain has ways to learning new information without forgetting old memories. Here, our focus is on incremental learning process which is similar to the process in human biological brains. Human memory is composed of three storage systems: 1)encoding, 2) long-term memory and 3) recall networks. Encoding system convert new information to a particular form for storage by paying attention Dorta. And, long-term memory has two main parts which are episodic and semantic memory and it acts as long-term warehouse and stores memories related to important personal events, feelings, silly facts, etc.,. Finally, the third network is recall system which is able to find information stored in our brain and "pull it out" efficiently.

Humans learn something new everyday and every moment which makes human brain to perform incremental learning process most of the time. To store information for long time in the brain, humans need to pay sufficient focus, time or attention on them at the time of learning. Furthermore,

in order not to forget the previously learned knowledge, humans must have sufficient practise, usage which involves revisiting to the stored memory Kirby (2013). If we compare DCNN with human brain, DCNN only has batch learning capability. It has no modules for information encoding, storing encoded information and recalling it for the purpose of practising or revisiting it later which are important for incremental learning process. In this paper, we integrate these missing modules to DCNN for incremental learning.

One concern of incremental learning system is the need to store training samples of categories trained previously. Previous methods such as state-of-the-art iCaRL Rebuffi et al. (2017b) and Roy et al. (2018) keep the old images for revisiting purpose. However, keeping original images long time in our CAA framework is not practical due to privacy and legal concerns. For example, it is not good to store negative images such as violence, accident scenes.

In this paper, we address two issues: 1) fulfilling the privacy and legal requirement 2) dealing with catastrophic forgetting problem with simulation of human brain system. The experimental results show that proposed solution is effective in addressing privacy and legal concerns as well as deling with catastrophic forgetting problem to maintain system performance especially for CAA framework.

## 2 RELATED WORK

Recently, there has been increased interest in incremental learning or lifelong learning and several approaches have been proposed to solve the problem. The approaches can be categorized into 3 main groups. The approaches in first group store the small subsets of training data from the previously learned tasks. The stored exemplars are used in training the network when adding the new tasks to alleviate the catastrophic forgetting problem. Example of such method is iCaRL Rebuffi et al. (2017b) which keeps a few exemplars to rehearse the previous tasks. Another recent method in this group is based on Gradient Episodic Memory (GEM) model Lopez-Paz & Ranzato (2017). However the usage of stored exemplars in GEM based approach is different from iCaRL Rebuffi et al. (2017b) which keeps the predictions of exemplars from past tasks invariant by means of distillation. GEM based approach define inequality constraints on loss to allow positive backward transfer for increasing performance on some preceding tasks. In second group, the approaches avoid storing any original training data from previously learned tasks. However, these approaches retain statistics of training samples such as means and standard deviations Kemker & Kanan (2018) of individual class distributions for tackling the catastrophic forgetting problem. Finally, the third group of algorithms totally avoid keeping any samples of previously learned tasks. Some of the approaches in this group tackle the catastrophic forgetting problem by dynamically expending the network for each new task Rusu et al. (2016), Rebuffi et al. (2017a), Yoon et al. (2018), Terekhov et al. (2015). Although these approaches are simple, they have scalability issues as the size of network grows with the increasing numbers of tasks. To avoid the scalability issue, some recent studies explore regularization methods over the network parameters Zenke et al. (2017), Kirkpatrick et al. or the study in Wu et al. investigates on generating synthesized images . Example of regularization method is Elastic Weight Consolidation (EWC) Zenke et al. (2017), Kirkpatrick et al., Lee et al. (2017), Liu et al. (2018), Chaudhry et al. (2018) based approach. EWC based approaches use a regularization term so that network parameters remain close to the parameters of the network trained for the previous classes. Similarly, Learning Without Forgetting (LWF) Li & Hoiem (2016) performs regularization on predictions instead of network weights.

Our proposed approach falls under the second group and the prior work in this category is Fear-Net Kemker & Kanan (2018) which is brain-inspired algorithm for incremental class learning. The FearNet system involves three brain-inspired sub-modules: 1) short-term memory storage for recently learned knowledge which is inspired by the hippocampal complex (HC) of human brain 2) long-term memory storage inspired by the medical prefrontal cortex(mPFC) and 3) a sub-module to choose the memory system between HC and mPFC to use for a particular example. Catastrophic forgetting problem is addressed by using pseudorehearsal which allows the network to revisit the previous memories during incremental training. Although FearNet uses HC model as short-term storage and the model is erased after the information is transferred to long-term storage(mPFC) network. mPFC stores class statistics for pseudorehearsal in the form of mean and covariance matrix for each class to generate new exemplars (pseudo-examples).

In human's brain, encoding is crucial step to creating new memory Mastin (2018). Encoding process convert newly perceived knowledge into a construct for storage to recall it later. During encoding process, human pay more attention especially for memorable events. This causes neurons to fire more frequently, making the event to be encoded as a memory. For example, emotional events tend to increase attention and leading it to unforgettable memories. Hence, the more intense the attention, the more effective the encoding and this results the storing acquired knowledge for long time. In this paper, we propose the brain inspired incremental learning system and investigate the effect of quality of encoded information on incremental learning. We integrate Convolutional AutoEncoder (CAE) in incremental learning process for encoding knowledge. We show that our brain inspired model is efficient and CAE is useful for encoding for revisiting the stored knowledge later.

The state-of-the-art system such as iCaRL Rebuffi et al. (2017b) requires to storing original training examples for all the previously learned classes, making it challenging for privacy and legal requirements especially for CAA framework which has limitations in storing original images (or exemplars) owned by others. To address privacy issues, the study in Wu et al. uses images generated by Generative Adversarial Networks (GAN) to replace the exemplar set of original training samples. However GANs have limitations on image quality and mode dropping. GAN based method works relatively well on simple images, however, it is not easy to get realistic scene images such as plane crash (for example) which we are interested. In our approach, we take care of privacy and legal by integrating CAE in our system.

## 3 DATASETS

Experiments are conducted on the following datasets.

**CIFAR-100** contains 100 image classes. Each class has 500 training images and 100 testing images. Following the class-incremental benchmark protocol in iCaRL [22] on this dataset, 100 classes are arranged in a fixed random order and come in as P parts. Each part is with C = 100/P classes. A multi-class classifier is built with the first part that contains C classes. Then this classifier is adapted to recognize all 100 classes.

**IMDB-CAA** We collect an IMage Database for Context Aware Advertisement (IMDB-CAA). Most of these images are negative scenes with violence, accidents, gambling etc. The database has about 12,000 color or gray-scale images with all images are manually labeled and irrelevant images are removed. The categories of the images in IMDB-CAA are listed in Table 1. Each category has 500 training images and 100 testing images.

Table 1: List of the image categories in IMDB-CAA database

| Categories | Categories | Categories | Categories |
| --- | --- | --- | --- |
| Ambulance | Gambling chip | Church | Mosque |
| Building on fire | Gambling cube | Cocaine or heroin | Plane |
| Car | Gun | Crashed plane | Police |
| Car in accident | Knives | Fire engine | Roulette |
| Children and junior student | Marijuana | Gambling card | Temple |

## 4 PROPOSED BRAIN INSPIRED INCREMENTAL LEARNING SYSTEM

In this paper, we investigate a strategy to simulate the encoding and storing information as in human brain learning process. By encoding the original images using Convolutional Auto-Encoder (CAE), the negative images become not easily viewable during storage. By using the incremental learning mechanism, only encoded information is kept for future learning once new data is available. In the following section we discuss about Convolutional Auto-Encoder (CAE) and presents its use to address privacy issues as well as to simulate human brain encoding mechanism.

## 4.1 Convolutional Auto-Encoder (CAE)

CAEs are very popular in deep learning research for unsupervised learning methods. CAEs can be used for extracting useful features from unlabelled data as well as for detecting and removing input redundancies to preserve only essential aspects of information for robust and discriminative representations Masci et al. (2011), Turchenko & Luczak (2015). Unsupervised training process of CAEs tends to avoid local minima and improves the network's performance and stability Erhan et al. (2010). As an auto-encoder, CAEs are based on encoder-decoder paradigm in which input data or image is mapped to a lower-dimensional space (referred to as encoder part) and then, encoded information is expended to regenerate the input data or image (referred to as decoder part).

As discussed above, CAEs have encoder and decoder parts which we need for simulating the human brain. Encoder part is useful for information encoding, storing encoded information. And, decoder part is for decoding stored information for practising or revisiting it in future. For DCNN to achieve the capabilities as those in human brain, we propose to integrate the auto-encoder, CAE, to DCNN. By doing so, we expect DCNN to achieve the capability of information encoding, storing, and practising or revisiting the stored information later in order to overcome the catastrophic forgetting problem of DCNN.

To overcome the catastrophic forgetting problem and address privacy requirements, we propose to use CAEs to encode the original images and keep encoded pseudo-examples instead of keeping original images. There are several advantages of using CAEs over GANs. These include 1) possibility of recalling or regenerating high quality images from stored encoded information for future revisiting. 2) There is freedom to control on the size of the pseoduexemplars by designing autoencoder as required. If accuracy is more important we can design the network for larger pseoduexemplars. However, if the storage is very limited, we can build the network for smaller pseoduexemplars with the certain expense of system performance.

One may argue that instead of using CAEs, we may use some encryption method. The focus in encryption and cryptography are security, encryption/decryption time, etc., Mahajan & Sachdeva (2013). It is common that security of the algorithm depends on the length of the key. The longer key length will always support to good security feature Oad et al. (2014). And, longer key length also means larger file size. Hence, most of the encrypted file is larger than before it is encrypted. However, our focus is on privacy, storage, performance and simplicity for easy scalability. Storage and performance should be able to be adjusted according to requirement of the application.

The CAE network involves 6 convolution layers and the first 3 layers are for encoder part and the last 3 layers are for decoder part. The numbers of filters in convolutional layers are set as {16, 8, 8, 8, 8, 16} respectively. The decoder part is a mirror of the encoder part. We use binary cross entropy as loss function and adaptive learning rate method for optimization function. We use subset of ImageNet database Deng et al. (2009) consisting of 150,000 images to train the CAE network. We set the heights and widths of the images as 224 to use as input for training the network. We use encoder part for encoding images and obtain pseudo-examples. Then, we employ decoder part to regenerate images from stored pseudo-examples. Figure 1 shows the comparisons of CAE regenerated images from different pseudo-exemplar sizes with original images and images generated by GAN.

In Figure 1, we gradually reduce the size of pseudo-examples. The reconstructed image becomes blur when we reduce the size. At the 16.6% reduced size we notice that edges of the object becomes sharper with increased noise. This phenomena can be explained by the findings in image enhancement using deep autoencoders. In image enhancement task, there is a trade-off between denoising capability and perceived sharpness of enhanced image Lore et al. (2016). The model with higher denoising capability generates smoother edges and generated images becomes less sharp which eventually lose structural information. It can be seen in $3^{rd}$ row from bottom in Figure 1 for 16.6% pseudo-exemplar size that object edges appears sharper at the cost of having more noise if we compare these images with reconstructed images of other pseudo-exemplar sizes. In fact, the characteristics such as sharper edges and textures are desirable properties especially for image classification with Convolutional Neural Network (CNN).

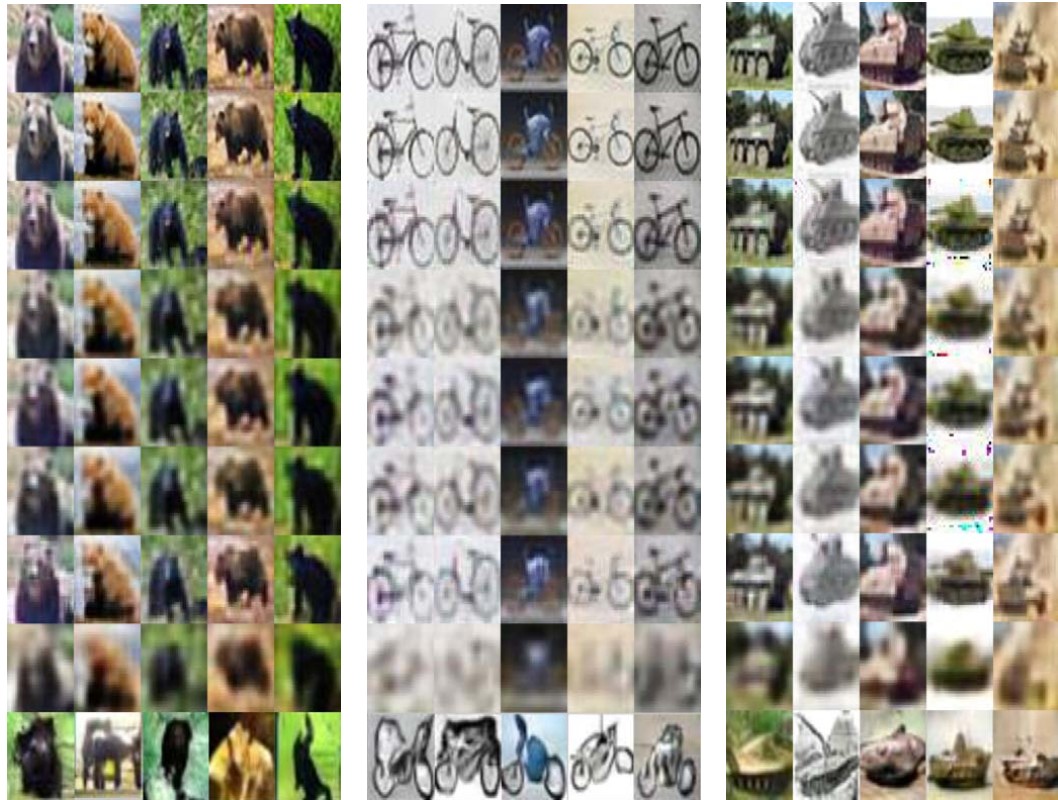

Figure 1: Illustrating CAE reconstructed images with various pseudo-exemplar sizes. $1^{st}$ row=real images, $2^{nd}$ to $8^{th}$ rows = CAE reconstructed images from pseudo-exemplar sizes of 200%, 100%, 67%, 50%, 33%, 16.7% and 4% of original real image size, last row = GAN generated images Wu et al.

## 4.2 CAE INTEGRATED PROPOSED INCREMENTAL LEARNING (IL) SYSTEM

The proposed IL system is developed based on ResNet152 He et al. (2015) pre-trained model. We integrate CAE to ResNet152 to simulate the processes of encoding and storing information as well as revisiting stored information as in human brain. CAE integrated proposed IL system is presented in Figure 2. We also integrate CAE on previous incremental learning method of iCaRL Rebuffi et al. (2017b). And, our focus is iCaRL to achieve capabilities as in human brain and to fulfill privacy requirements.

## 5 EXPERIMENTS

We investigate the effect on quality of encoded information in our brain inspired incremental learning system by conducting a systematic series of experiments. We conduct experiments on CIFAR-100 dataset. Firstly, we investigate effect of integrating CAE to state-of-the-art iCaRL system in terms of storage size. Secondly, we observe the performance of our proposed IL system on quality of encoded information. Thirdly, we compare performance of our proposed system with state-of-the-art iCaRL as well as other recent systems. Finally, we present experiments on our IMDB-CAA dataset. As in Rebuffi et al. (2017b), the evaluation measure is the standard multi-class accuracy on the test set.

Original size of CIFAR-100 is only 32x32x3 pixels while IMDB-CAA is at least 300x300x3 pixels. The image sizes of the two datasets are largely different. Unless otherwise stated, we employ CAE with the architecture presented in Section 4.1 with the pseudo-exemplar size of 28x28x8 in our experiments.

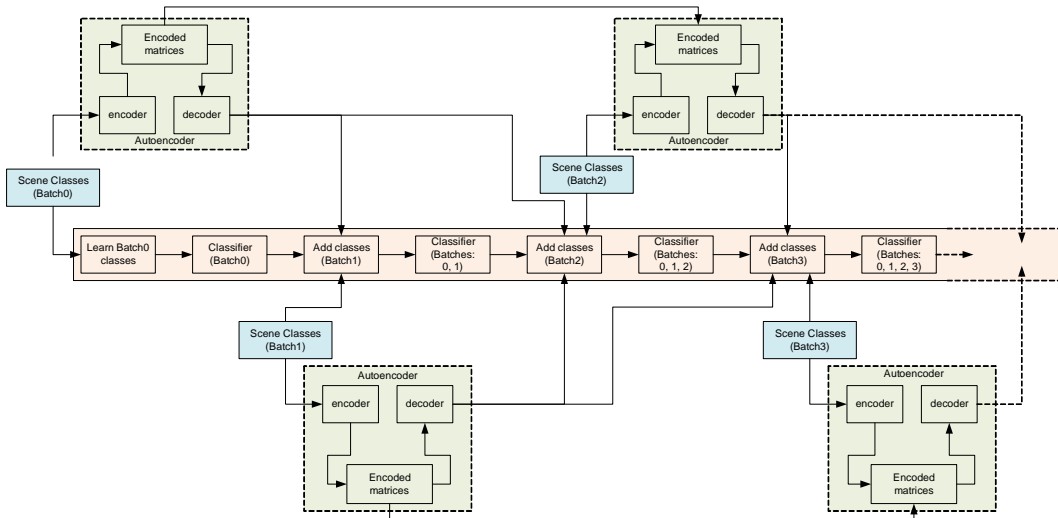

Figure 2: CAE integrated proposed IL system with human brain inspired information encoding, storing and recalling information and revisiting processes.

## 5.1 EXPERIMENTS ON CIFAR-100 DATASET

We conduct experiments using public dataset CIFAR-100 which is used in state-of-the-art iCaRL Rebuffi et al. (2017b) and GAN based Wu et al. incremental learning systems. The following is experiments on the effect of different pseudo-exemplar set size on CAE integrated iCaRL system.

### 5.1.1 EFFECT OF PSEUDO-EXEMPLAR SET SIZE ON CAE INTEGRATED ICARL SYSTEM

Original iCaRL system keeps real images (referred to as exemplars) for future revisiting. Here, we encode the images before we keep them. We integrate CAE to iCaRL system (with P=10, C=10) to conduct experiments. Quality of encoding process has effect on system performance. One way to improve quality of encoded information is to store as many examples as possible to cover the distribution of a particular image class. We gradually increase the numbers of pseudo-examples in our experiments. We use pseudo-examples only for revisiting past tasks. As for training new classes and testing, we use original samples. Table 2 shows the classification accuracies on different set sizes of pseudo-examples.

Table 2: Average Accuracies (%) of incremental learning system with different sizes of pseudo-exemplar sets with CAE integrated to iCaRL system on CIFAR-100 dataset.

| Pseudo-exemplar set size | Accuracies |
|---|---|
| 2000 | 45.2 |
| 4000 | 49.33 |
| 6000 | 50.45 |
| 8000 | 52.91 |
| 10000 | 53.59 |
| 12000 | 55.18 |
| Full exemplar set | 56.7 |

In Table 2, pseudo-exemplar set size 2000 means that system keeps maximum 2000 samples for all past tasks at all time as in iCaRL system Rebuffi et al. (2017b). The results show that system performance improves with increasing pseudo-exemplar set size. We also use full pseudo-exemplar set to observe the highest performance that CAE integrated iCaRL can achieve. As can be seen in last row of Table 2, when we use full exemplar set, accuracy increase to 56.7%. In the following, we observe the effect of pseudo-exemplar sizes on CAE integrated proposed IL system.

### 5.1.2 EFFECT OF DIFFERENT PSEUDO-EXEMPLAR SIZES ON CAE INTEGRATED PROPOSED IL SYSTEM

The size of pseudo-example has direct impact on storage. The smaller size is preferable especially for devices with limited storage. We conduct experiments to observe the effect on different sizes of pseudo-examples and system performance. We modify the layers of CAE presented in Section 4.1 to obtain different pseudo-exemplar sizes. The first 3 columns of Table 3 shows the experimental results. In the first column, 'Same' means the size of pseudo-example is the same as original real image size. And, 67% means the size of a pseudo-example is 67% of the size of corresponding real image in the CIFAR-100 dataset. We use all 500 pseudo-exemplars to train each classes. The system is tested on real images as well as on pseudo-samples. Training is done using only pseudo-examples for all old and new tasks.

Table 3: Average Accuracies (%) CAE integrated proposed IL system and original iCaRL system on different sizes of pseudo-example. Test on real images and pseudo-samples of CIFAR-100 test set.

| Proposed system | | | iCaRL | |
|---|---|---|---|---|
| Pseudo-exemplar size | Real | Pseudo-exemplar | Exemplar set size | Real |
| Same | 68.44 | 65.96 | Same | 61.89 |
| 67% | 47.47 | 49.65 | 67% | 57.37 |
| 50% | 44.82 | 49.56 | 50% | 57.19 |
| 33.3% | 42.6 | 47.1 | 33.3% | 56.18 |
| 16.6% | 51.5 | 50.66 | 16.6% | 52.64 |

We achieve relatively high accuracies for both test conditions: pseudo-samples and real images when pseudo-exemplar size is the same as real image size. However, when we reduce the pseudo-exemplar size to 67% performance starts to degrade. The quality of reconstructed images from pseudo-examples is not as good as real images as can be seen in Figure 1. With decreasing pseudo-exemplar size, the reconstructed images becomes more and more blur and edges appear less and less sharp. However, when we further reduce the pseudo-exemplar size to 16.6% system performance improves as we can see in the last row of Table 3 for both pseudo-sample and real test images. As we have discussed in Section 4.1, pseudo-examples with the size of 16.6% gives reconstructed images with sharper edges at the expense of increasing noise. And, shapes and colors of background start to fade (or blend) and object standouts from the background. This makes feature maps of the CNN classifier easy to detect the edges of the object and we are able to achieve the higher classification performance with much less storage requirement. In the following section we compare our proposed system with recent studies.

### 5.1.3 COMPARING PROPOSED IL SYSTEM WITH STATE-OF-THE-ART SYSTEMS

We compare our system with the state-of-the-art iCaRL system Rebuffi et al. (2017b), GAN based system Wu et al., FearNet system Kemker & Kanan (2018). Firstly, we compare our proposed system with iCaRL for same storage requirements. We conduct experiments with iCaRL on different exemplar storage settings and results are reported in the last $5^{th}$ column of Table 3. The $4^{th}$ column of the table shows the storage requirement of iCaRL system. For example, 67% means 67% of the training set is stored as exemplar set. As we can see, our proposed system perform better than iCaRL when the pseudo-exemplar size is the same is original image size. For less storage requirement option, our proposed system achieve very similar performance as in iCaRL for the storage requirement of 16.6%. Please note that iCaRL stores real exemplars while our system stores pseudo-examples and hence, our approach full fill the privacy and legal requirements. To compare our system with GAN based system Wu et al., FearNet system Kemker & Kanan (2018), we repeat the results of Table 3 together with the system performances of the state-of-the-art systems in Table 4.

As we can see in the table, our system achieves 68.44% accuracy which is the highest among all systems. This setup is useful especially for our application in CAA framework in which storage is less concern than privacy. And, the proposed system also has the option with less storage requirement for reasonably high accuracy as can be seen in the last row of the table for 16.6% pseudo-exemplar size. As mentioned previously, human pays intense attention for better encoding of knowledge for

Table 4: Comparing performance of proposed IL system with other incremental learning approaches on CIFAR-100 test set. P=10 and C=10 for all iCaRL systems.

| Approach | Exemplar set | Accuracies |
|---|---|---|
| GAN | | 54.9 |
| FearNet | | 66.2 |
| iCaRL | Same | 61.89 |
| iCaRL | 16.6% | 52.64 |
| Proposed system | Same | 68.44 |
| Proposed system | 16.6% | 51.5 |

unforgettable memory. Similarly, in our proposed system, quality of encoded statistics is directly related to size of the pseudo-example. However, smaller pseudo-exemplar size does not necessarily means having incomplete information.

Another advantage of the proposed system is that we do not need to train individual CAE for each image class. We prepare only one CAE, train it only for once and, then, it can be used for life time of IL system. GAN based method needs training image generator for each image category. And, storage requirements of each GAN model could be much larger than that of the whole original training set especially for very small images such as CIFAR-100.

## 5.2 EXPERIMENTS ON IMDB-CAA DATASET

Finally, we conduct experiments using IMDB-CAA dataset on our proposed IL system. Original size of the images in this data set is much larger than CIFAR-100 dataset. In the following we conduct experiments using different pseudo-exemplar sizes. We reduce the pseudo-exemplar sizes up to 4% of original image. Table 5 shows the results.

Table 5: Average Accuracies (%) of proposed IL system with different pseudo-exemplar sizes. Test on both pseudo-samples and real images of IMDB-CAA test set.

| Pseudo-exemplar size | Real | Pseudo-sample |
|---|---|---|
| Same | 82.1 | 84.1 |
| 66.7% | 51.9 | 81.3 |
| 50% | 37.3 | 74.9 |
| 33.3% | 53.4 | 78.85 |
| 16.7% | 48.8 | 81.2 |
| 4 % | 37.3 | 74.95 |

For this IMDB-CAA dataset, the system performance with the offline batch training on real images is 86.6%. We use ResNet152 pre-trained model and fine-tuned with the training set in our offline batch training. As we can see in the $2^{nd}$ row of Table 5, we can achieve the classification performance which is very close to offline batch training. Even at the pseudo-exemplar size of 4%, system performance is reasonably high. As in CIFAR-100 data set, IMDB-CAA also shows the best pseudo-statistics for 16.6% size which gives 81.2% accuracy and the performance is comparable to offline batch training performance with original images. When we examine the images, we notice the same image characteristics, sharp edges, as in CIFAR-100.

## 5.3 CONCLUSIONS

In this paper, we presents a strategy to simulate brain inspired information encoding and storing for incremental learning system. With this strategy, we also address the privacy and legal issues which is required for incremental learning in our Context Aware Advertisement (CAA) framework. Experimental results show that proposed strategy is simple, efficient and easily scalable. A single system is able to provide various options on system performance which is better than state-of-the-art to the performance which is reasonably high with very little storage for both CIFAR-100 and IMDB-CAA datasets.

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
