# OpenReview forum: "ON THE USE OF CONVOLUTIONAL AUTO-ENCODER FOR INCREMENTAL CLASSIFIER LEARNING IN CONTEXT AWARE ADVERTISEMENT"
_ICLR.cc/2019/Conference_

### Official Review · AnonReviewer3 · 2018-11-02
**The paper adresses the problem of incremental learning when data from new classes are available as a stream and one wants to be able to update to learn new observed classes without forgetting the older ones.  This is a relevant problem but the paper strongly lacks precision about the proposed method that looks like an incremental work with limited innovation but that the reader can only try to guess.**

**Rating:** 3
**Confidence:** 4

**Review:**

The paper adresses the problem of incremental learning when data from new classes are available as a stream and one wants to be able to update to learn new observed classes without forgetting the older ones. There is a budget issue here and one does not want to just keep the whole training set of all known previously observed classes but rather one wants to consider a maximum memory budget allowed to store what is necessary for an optimal incremental learning (typical examples, statistics etc). There is also a privacy issue preventing from storing original training samples.

This is a relevant problem that is has gain interest in the last few years. It is related to topics such as few shot learning and meta few shot learning (with respect ti the number of examples per class that are kept, which is limited) and somehow to budget learning . Yet these topics and associated references are surprisingly not evoked in the text.

The paper is rather well written but it strongly lacks precision about the proposed method. A description of the ICARL state of the art method is missing and would have been mandatory since the proposed work appears to build on iCARL method. Actually the description of the method is very short since the dedicated section (§4) is mainly used to describe a rather standard convolutional auto encoder architecture. At the end one tries to guess what the proposed method consists in. As far as i understand it is based on iCARL method where selected examples of past observed classes are not stored as is but in their encoded form (by the convolutional autoencoder). At the end my understanding of the proposed approach is that it consists in an incremental progress of a state of the art method, then an incremental work with limited innovation.

By the way i am not sure of the meaning of pseudo exemplar as used in the proposed method. Are these drawn following a distribution computed on training samples ? Or are these pseudo exemplar because you use reconstructed samples from encodings (by the CAE).

When looking at experimental results the proposed method seem to bring some benefit but it does not look fully convincing. As written in the paper the proposed system outperforms iCARL in case the examples are encoded in the same dimension as original examples (hence no benefit on the storage side) but reaches similar performance when using less storage capacity.

---

### Official Review · AnonReviewer2 · 2018-11-03
**Several misleading claims**

**Rating:** 4
**Confidence:** 4

**Review:**

This paper describes a system for classifying  images  displayed  on  the  websites  by  using an incremental  learning  classifier with Deep Convolutional Neural Network to be used in context aware advertisement.

This is more of an application paper which is not the focus of a venue like ICLR. Further the paper makes several misleading claims. They claim that their system is inspired by the human brain while providing scant evidence to prove that claim (Unless we take it in a very broad sense to mean that neural networks resemble the human brain). Also they propose a convolutional autoencoder as a kind of an encryption method to store images to alleviate privacy and legal concerns. This is not a good idea because encoding an image using a convnet is not a substitute for encryption. In fact any image hence encoded can be decoded easily to reveal the original contents of the image. Overall this paper is not appropriate for ICLR in its current form.

---

### Official Review · AnonReviewer1 · 2018-11-04
**The paper deals with a significant topic, that of incremental classifier learning, but has a limited novelty and many unclear points**

**Rating:** 5
**Confidence:** 5

**Review:**

The paper extends an existing incremental learning method, mainly introducing the latent representations of an autoencoder instead of the original images. It includes a lot of hype in that it simulates the human brain -  because it is based on the iCaRL & Fear Net formulation - and that it fulfils the privacy and legal requirements - because it stores and uses the auto-encoder representations instead of the images.

Specific comments:

- The title of the paper defines its topic to be context aware advertisement, whereas the main results and all comparisons are made on the CIFAR dataset. Only the last Table (5) provides the performance on the IMDB-CAA dataset, without any detailed analysis of the experiments.

- The results in Table 3 are quite strange: the presented approach starts by outperforming iCaRL method, but then deteriorates very fast wrt size and is much lower than the original method, with no justification on this. Some improvement is shown in size 16.6% without, again, any logical explanation provided.

- Section 4.2 does not provide any detail of the integration resulting in the presented system; Fig.2 does not provide a clear description either.

- Language improvement is required in the experimental sections.

-

---

### Meta-Review · Area_Chair1 · 2018-12-13
**interesting problem, but contribution is hard to assess due to lack of details and clarity**

**Confidence:** 4
**Recommendation:** Reject

**Metareview:**

1. Describe the strengths of the paper.  As pointed out by the reviewers and based on your expert opinion.

The paper tackles an interesting and relevant problem for ICLR: incremental classifier learning applied to image data streams.

2. Describe the weaknesses of the paper. As pointed out by the reviewers and based on your expert opinion. Be sure to indicate which weaknesses are seen as salient for the decision (i.e., potential critical flaws), as opposed to weaknesses that the authors can likely fix in a revision.

- The proposed method is not clearly explained and not reproducible. In particular the contribution on top of the baseline iCaRL method is unclear. It seems to be mainly the use of CAE which is a minor change.
- The experimental comparisons are incomplete. For example, in Table 4 the authors don't discuss the storage requirements of GAN and FearNet baselines.
- The authors state that one of their main contributions is fullfilling privacy and legal requirements. They claim this is done by using CAEs which generate image embeddings that they store rather than the original images. However it's quite well known that a lot of data about the original images can be recovered from such embeddings (e.g. Dosovitskiy & Brox. "Inverting visual representations with convolutional networks." CVPR 2016.).
These concerns all impacted the final decision.

3. Discuss any major points of contention. As raised by the authors or reviewers in the discussion, and how these might have influenced the decision. If the authors provide a rebuttal to a potential reviewer concern, it’s a good idea to acknowledge this and note whether it influenced the final decision or not. This makes sure that author responses are addressed adequately.

There were no major points of contention and no author feedback.

4. If consensus was reached, say so. Otherwise, explain what the source of reviewer disagreement was and why the decision on the paper aligns with one set of reviewers or another.

The reviewers reached a consensus that the paper should be rejected.